# Impact of NG-Test CTX-M MULTI Immunochromatographic Assay on Antimicrobial Management of *Escherichia coli* Bloodstream Infections

**DOI:** 10.3390/antibiotics12030473

**Published:** 2023-02-27

**Authors:** Matteo Boattini, Gabriele Bianco, Davide Ghibaudo, Sara Comini, Silvia Corcione, Rossana Cavallo, Francesco Giuseppe De Rosa, Cristina Costa

**Affiliations:** 1Microbiology and Virology Unit, University Hospital Città della Salute e della Scienza di Torino, 10126 Turin, Italy; 2Department of Public Health and Paediatrics, University of Torino, 10126 Turin, Italy; 3Department of Medical Sciences, Infectious Diseases, University of Turin, 10126 Turin, Italy; 4Unit of Infectious Diseases, Cardinal Massaia, 14100 Asti, Italy

**Keywords:** *Escherichia coli*, ESBL, rapid diagnostics, CTX-M, bloodstream infection, blood culture, immunochromatographic assay

## Abstract

Rapid detection of extended-spectrum-β-lactamase (ESBL) is of paramount importance to accelerate clinical decision-making, optimize antibiotic treatment, and implement adequate infection control measures. This study was aimed at assessing the impact of direct detection of CTX-M ESBL-producers on antimicrobial management of *Escherichia coli* bloodstream infections over a 2-year period. This study included all *E. coli* bloodstream infection (BSI) events that were serially processed through a rapid workflow with communication to the clinicians of direct detection of CTX-M ESBL-producers and conventional culture-based workflow. Antimicrobial management was retrospectively analyzed to assess the contribution of the rapid test result. A total of 199 *E. coli* BSI events with a report of direct detection of CTX-M ESBL production results were included. Of these, 33.7% (n = 67) and 66.3% (n = 132) were reported as positive and negative CTX-M producers, respectively. Detection of CTX-M positive results induced more antibiotic therapy modifications (mainly towards carbapenem-containing regimens, *p* < 0.01), and antimicrobial susceptibility testing results of CTX-M ESBL-producing *E. coli* isolates induced more antibiotic escalations towards carbapenem-containing regimens (*p* < 0.01). Direct detection of CTX-M ESBL-producing *E. coli* resulted in a remarkable rate of antibiotic optimizations on the same day of blood culture processing. Observing antibiotic management following the availability of antimicrobial susceptibility testing results, additional early optimizations in escalation could probably have been made if the rapid test data had been used. Detection of CTX-M negative results resulted in few therapeutic changes, which could have probably been higher, integrating epidemiological and clinical data.

## 1. Introduction

In recent years several rapid non-molecular tests for the detection of the main antibiotic resistance enzymes in Gram-negative bacteria have been developed and introduced into the routine of many laboratories [1,2,3,4,5,6,7,8,9,10,11,12]. They have also been favorably evaluated directly from positive blood cultures (BCs) with the purpose of providing results on the same day of sample processing, at least 24 h earlier than conventional susceptibility testing [11]. Rapid detection of the main β-lactamases is of paramount importance to accelerate clinical decision-making, optimize antibiotic treatment, and implement adequate infection control measures [13]. Additionally, rapid characterization of these enzymes can help to guide therapy, as the type of β-lactamase confers different resistance spectra to carbapenems and novel β-lactam/β-lactamase inhibitor combinations. Routine infectious disease bedside consultation planned within antimicrobial stewardship programs on rapid susceptibility testing results was reported to change antimicrobial treatment in more than 50% of cases [14]. However, since bedside consultations are not routinely available in many hospitals, written diagnosis–treatment recommendations on microbiological test reports have also been implemented with no effect on mortality [15]. Antimicrobial resistance at the hospital level is an issue that is unlikely to be tackled if delegated to infectious disease and clinical microbiology specialists alone, as knowledge of local epidemiology, antimicrobial prescribing, as well as interpretation of susceptibility results affects all hospitalists. In this regard, it is not known how rapid test results on the detection of the main antibiotic resistance enzymes can impact antibiotic consumption and clinicians’ confidence to change antibiotic therapy, especially to de-escalation, since other resistance mechanisms non-detectable by the rapid test used could be present.

Extended-spectrum-β-lactamase (ESBL)-producing Enterobacterales (EB) infections represent a worldwide issue concerning public health, especially given their association with poor outcomes, growing community-onset, and high ecological treatment cost [16,17,18,19]. ESBL enzymes are, in fact, the main actors in EB in conferring resistance to penicillins, cephalosporins, and aztreonam. Third-generation cephalosporin-resistant *Escherichia coli* and *Klebsiella pneumoniae* have been recently reported to contribute to high numbers of attributable deaths and disability-adjusted life-years per 100,000 individuals [20]. Moreover, from both therapeutic and ecological points of view, the burden of ESBL-producing EB infections is very heavy since carbapenems are the proven treatment option [19].

Although the ESBL family is heterogeneous, the global pandemic of plasmids carrying CTX-M type genes, which started mainly in the 2000s, is the main driver of ESBL dissemination in EB and has replaced other ESBL enzymes (i.e., mostly TEM, SHV derivatives) [21]. A recent survey, as part of the International Network for Optimal Resistance Monitoring (INFORM) global surveillance program on EB and *Pseudomonas aeruginosa* isolates collected from 18 European countries, reported 18.5% of ESBL-producers in *E. coli* isolates, CTX-M-type enzymes being the most frequently detected [22]. Similarly, 35.5% of *K. pneumoniae* isolates were ESBL-producers, and CTX-M-15 enzymes comprised more than 70% of ESBLs detected. Of note, an elevated incidence of SHV-type ESBL-producing *K. pneumoniae* was found in Southern Europe (17%), reaching 64% of those identified in Greece [22]. The recent introduction of lateral flow immune assays into the market has brought about a real revolution in the field of antimicrobial resistance detection, as it has given every laboratory the opportunity to equip itself with reliable tools without the need to have technical expertise or expensive instrumentation [23]. The lateral flow NG-Test CTX-M MULTI assay (NG Biotech, Guipry, France) exploits monoclonal antibodies specific for CTX-M variants belonging to group 1 (including CTX-M-15), group 2, group 8, group 9 (including CTX-M-14), and group 25. It detects CTX-M-type ESBLs from both bacterial cultures and pellets, providing results in <15 min without discriminating CTX-M variant or subgroup, and requires no specific storage constraints, minimal hands-on time, and no additional equipment [8,10,11,23]. Given the ESBLs epidemiological context and with the aim of both providing reliable and rapid microbiological results and maximizing cost-effectiveness, the NG-Test CTX-M MULTI assay (NG Biotech, Guipry, France) has been implemented in the BC workflow of our laboratory since November 2019 [11].

This study was aimed at assessing the impact on the antimicrobial prescription of direct detection of CTX-M ESBL-producers in *E. coli*-positive BCs in an Italian University hospital over a 2-year period.

## 2. Results

One-hundred ninety-nine *E. coli* BSI events with a report of direct detection of CTX-M ESBL production results were included in the study. Of these, 33.7% (n = 67) and 66.3% (n = 132) were reported as positive and negative CTX-M producers, respectively (Table 1). Comparing the antimicrobial resistance patterns, CTX-M ESBL-producing *E. coli* were more resistant to ceftazidime (*p* < 0.01), cefotaxime (*p* < 0.01), cefepime (*p* < 0.01), ceftolozane/tazobactam (*p* = 0.04), aminoglycosides (*p* < 0.01), fluoroquinolones (*p* < 0.01), colistin (*p* = 0.04), and sulfamethoxazole/trimethoprim (*p* < 0.01) than *E. coli* isolates with CTX-M negative results. No statistically significant difference was found for ceftazidime/avibactam, piperacillin/tazobactam, ertapenem, meropenem, and imipenem. Of note, among *E. coli* isolates with CTX-M negative results, 3.8% (n = 5) and 1.5% (n = 2) were ESBL-producers other-than-CTX-M-types detectable by the lateral flow immunoassay used in the study and AmpC-producers, respectively.

From the analysis of antimicrobial clinical attitude according to direct CTX-M results (Table 2) emerged (1) no statistically significant differences in empirical antibiotic therapy except that patients with CTX-M negative result were treated with more active empirical therapy (*p* < 0.01); (2) direct detection of CTX-M positive result induced more antibiotic therapy modifications, mainly toward carbapenem-containing regimens and less toward 3rd generation cephalosporin- and piperacillin/tazobactam-containing regimens (*p* < 0.01); (3) antimicrobial susceptibility testing results of CTX-M ESBL-producing *E. coli* isolates induced more antibiotic escalations toward carbapenem-containing regimens (*p* < 0.01); (4) antimicrobial susceptibility testing results of *E. coli* isolates with CTX-M negative result induced more antibiotic de-escalations toward 3rd–4th generation cephalosporin-containing regimens (*p* < 0.01).

## 3. Discussion

Rapid tests for the detection of the main resistance enzymes in Gram-negative bacteria have been developed with the final objectives of implementing efficient infection control measures and identifying resistance mechanisms so that the most appropriate antibiotic treatment can be started. *E. coli* belongs to the small number of Gram-negative bacteria with significant clinical impact and is, therefore, one of the most studied [24]. This study reported a real-life experience assessing the impact on clinicians’ confidence and antimicrobial prescription of a newly introduced diagnosis of direct detection of CTX-M ESBL-producers in *E. coli*-positive BCs in a hospital in which carbapenem-sparing strategies were implemented during the last years, mainly as educational interventions. The lateral flow NG-Test CTX-M MULTI assay showed to be well adapted to the Italian epidemiology since only 3.8% of *E. coli* isolates that tested negative to CTX-M expressed ESBL enzymes other than CTX-M-types. Direct detection of CTX-M positive result allowed optimizing more antibiotic therapies (mainly towards carbapenem-containing regimens) on the same day of BC processing than CTX-M negatives. However, in more than 20% of patients with direct detection of CTX-M positive result, the antibiotic escalation was only performed after the antimicrobial susceptibility testing results were available, carbapenem-containing regimens being the most prescribed (79%). Direct detection of CTX-M negative results induced very few changes in therapy (13.6%). Antimicrobial susceptibility testing results obviously allowed more antibiotic de-escalations (mainly to 3rd–4th generation cephalosporin-containing regimens) in patients suffering from BSI caused by *E. coli* with CTX-M negative result.

Implementation of rapid point-of-care diagnostic tests for antimicrobial resistance markers is considered mandatory to achieve efficient infection control measures, identification of resistance mechanisms, and appropriate antibiotic therapy [23]. The choice of the most appropriate rapid diagnostic workflow from BCs should consider laboratory organization as well as local epidemiology of resistance mechanisms [12,13]. Several approaches have been evaluated and shown to provide very accurate results that take into account the presence of multiple mechanisms of resistance. The whole-genome sequencing-based approach provided reliable results as those obtained by phenotypic AST [25,26]. Rapid AST was also favorably evaluated from BCs, conferring to the method greater potential, especially for antimicrobial de-escalation interventions [27]. However, both of these approaches are only rarely implemented at present for different logistical reasons. To the best of our knowledge, our study was the first that sought to quantify the degree to which clinicians received the microbiological report with CTX-M positive or negative results, verified by a cheap and easy-to-use rapid immunochromatographic testing directly from the vial in a dedicated workflow to *E. coli*. The decision to communicate by laboratory information system with an extremely simple text was made to (1) leave no room for interpretation or attempted interference about the change of antibiotic therapy given our unawareness of patients’ clinical condition; (2) reach patients suffering from BSI caused by *E. coli* with CTX-M negative result and multi-susceptible phenotype, who are rarely included in antimicrobial stewardship programs. Our results highlighted that direct detection of CTX-M ESBL-producing *E. coli* persuaded clinicians to escalate antibiotic therapy to a significant but unsatisfactory extent, given the considerable number of carbapenem-containing prescriptions following antimicrobial susceptibility testing results. We speculated that this finding might be related to either a lack of confidence in rapid test results or “CTX-M” nomenclature, highlighting the need for multifaceted interventions targeting all the prescribers to inform about reliability [8,10,11] and operating principles of the rapid tests for the detection of the main resistance enzymes in Gram-negative bacteria. Conversely, communication of CTX-M negative results resulted in very few changes in therapy, mainly antibiotic therapy introduction and escalation, probably due to *E. coli* identification and specific clinical considerations, respectively. Although antibiotic de-escalation could present a dark side, reducing antimicrobial exposure is considered essential [28]. In our study, antimicrobial resistance patterns of *E. coli* with CTX-M positive and negative results were very different, being the latter resistance rates to 3rd generation cephalosporins, piperacillin-tazobactam, and fluoroquinolones very lower (<5%, <8% and <19%, respectively). This finding should prompt us to consider that the knowledge of local epidemiology (low number of AmpC/ESBL-producing *E. coli*) together with the knowledge of the patient’s clinical condition might set the field on which antibiotic de-escalation may be implemented from the result of the rapid test.

The attempt to quantify the immediate benefits and limitations provided by the implementation of a rapid test on antimicrobial prescriptions of septic patients is certainly a strength of our study. The lack of knowledge of both clinical contexts (e.g., severity of BSI, source of infection, source control rate, use of clinical scores such as the INCREMENT-ESBL score), which might have influenced the choice of antibiotic escalation or de-escalation and patients’ outcomes were the main limitations of this study.

## 4. Conclusions

The provision of microbiological test report with the diagnosis of direct detection of CTX-M ESBL-producing *E. coli* resulted in a remarkable rate of antibiotic optimizations on the same day of BC processing. Moreover, observing antibiotic management following the availability of antimicrobial susceptibility testing results, this real-life study suggests that additional early optimizations in escalation could probably have been made if the rapid test data had been used. Direct detection of CTX-M negative results resulted in few therapeutic changes, which could have probably been in greater numbers integrating epidemiological and clinical data. Multifaceted interventions for all the prescribers to illustrate the potential of rapid tests for the detection of the main resistance enzymes in Gram-negative bacteria should be considered part of the implementation cost. Further studies on the impact on antibiotic prescribing of the written prediction of antibiotic susceptibility according to rapid resistance phenotype are desirable to establish a horizon in which to align clinical and ecological outcomes.

## 5. Materials and Methods

### 5.1. Study Design

This study was performed at the University Hospital Città della Salute e della Scienza di Torino, a 1900-bed tertiary care teaching hospital in Turin, northwestern Italy, from July 2020 to June 2022. This study included all *E. coli* positive BCs deemed representative of a single bloodstream infection (BSI) event that were serially processed through two microbiological diagnostics: (1) rapid workflow with communication of direct detection of CTX-M ESBL-producers report on the laboratory information system; (2) conventional culture-based workflow. The study considered only one BC bottle per patient/BSI event while excluding those collected from patients with previous Gram-negative BSI within the previous 15 days. Comparison of antimicrobial resistance patterns of *E. coli* isolates according to rapid phenotypic characterization and antimicrobial management (empirical antibiotic therapy, therapeutic modifications after the rapid diagnostic result, therapeutic modifications after the availability of conventional susceptibility testing results) were retrospectively analyzed to assess the contribution of the rapid test result on antimicrobial management.

### 5.2. Rapid Blood Culture Workflow: Direct Detection of CTX-M ESBL-Producing E. coli

The positivity of all BC bottles during the study was detected using the BactAlert Virtuo instrument (Marcy l’Ètoile, France). Positive BC bottles showing Gram-negative bacteria on microscopic examination were processed through the rapid diagnostic workflow as indicated below. (1) Recovery of the bacterial pellet by Rapid MBT Sepsityper^®^ IVD Kit (Bruker Daltonik, Bremen, Germany); (2) MALDI-TOF MS analysis by Bruker Microflex LT mass spectrometer; (3) NG-Test Carba 5 (NG Biotech, Guipry, France) for the detection of the main carbapenemases (KPC, VIM, NDM, OXA-48-like) and NG-Test CTX-M MULTI assays on *E.* coli BC bottles using the remaining pellet, as previously described [11]; (4) In case of NG-Test Carba 5 negative result, communication to the clinicians through the laboratory information system of a standardized report with one of the following two options: (a) Species identification: *E. coli*, positive CTX-M ESBL-producer; (b) Species identification: *E. coli*, negative CTX-M ESBL-producer.

### 5.3. Conventional Blood Cultures Routine

The Microbiology and Virology Unit is part of the Azienda Ospedaliera Universitaria “Città della Salute e della Scienza di Torino”. Clinical samples are accepted seven days a week from 08:00 to 20:00. Outside these hours, routine clinical samples reaching the laboratory are processed the next day, while those deemed essential for the therapeutic management of patients are processed by an emergency laboratory technician and validated by a microbiologist. During the study period, BACT/ALERT^®^ FA Plus aerobic BACT/ALERT^®^, FN Plus anaerobic, and BACT/ALERT^®^ PF Plus pediatric bottles (bioMérieux, Marcy l’Ètoile, France) were used to process BCs and incubated in the BACT/ALERT^®^ Virtuo^®^ (bioMérieux). BC bottles flagged positive by the BACT/ALERT^®^ Virtuo^®^ underwent subculture on solid media at 36 ± 1 °C and proper atmosphere conditions and slides preparation using WASPLab^®^ instrument (Copan, Brescia, Italy). The automated stainer Aerospray^®^ (ELITechGroup, Turin, Italy) was used to perform Gram staining of slides. Microbial identification was performed on overnight subcultures with MALDI-TOF MS following the manufacturer’s instructions. Susceptibility of non-fastidious Gram-negative isolates to several antibiotics (cefotaxime, ceftazidime, cefepime, piperacillin/tazobactam, ceftolozane/tazobactam, ceftazidime/avibactam, meropenem, imipenem, ertapenem, gentamicin, amikacin, colistin, trimethoprim/sulfamethoxazole, ciprofloxacin, and levofloxacin) was tested with an automated microdilution assay (Panel NMDR on automated Microscan WalkAway 96 Plus System, Beckman Coulter, Switzerland) according to the manufacturer’s instructions. EUCAST guidelines (https://www.eucast.org (accessed on 1 January 2023)) were used to identify ESBL- and carbapenemase-producing Enterobacterales strains, and confirmatory tests for resistance mechanisms were performed once the conventional antimicrobial susceptibility testing results became available. The phenotypic test included in the NMDR microdilution panel, based on synergy of β-lactamases inhibitor clavulanic acid on cefotaxime and ceftazidime MICs, was used to detect ESBLs. Multiplex real-time polymerase chain reaction assay specific for blaCTX-M-like genes (ESBL ELITe MGB Kits, ELITechGroup Molecular Diagnostics, Turin, Italy) was used on isolates that tested positive by phenotypic test for ESBL detection. Eazyplex^®^ SuperBug AmpC (AmplexDiagnostics GmbH, Gars am Inn, Germany) was used for the detection of AmpC β-lactamases (ACC, CMY-II, DHA, MOX) on isolates with ceftazidime and/or cefotaxime MIC > 1 mg/L that tested negative by phenotypic test for ESBL detection. The genotypic assay Xpert Carba-R on the GeneXpert platform (Cepheid, Sunnyvale, CA, USA) was used to investigate the main carbapenemase genes in EB (blaKPC, blaNDM, blaVIM, blaIMP, and blaOXA-48-like) when meropenem and/or ertapenem MICs were >0.12 mg/L. Microbial identification and susceptibility results were promptly communicated to clinicians through the laboratory information system.

### 5.4. Definitions

Antibiotic therapy was deemed empirical when administered during the period prior to the receipt of conventional BC results. Combination therapy refers to the use of two or more antibiotics. Empirical antibiotic therapy was deemed active when a causative bacterial strain was susceptible in vitro to at least one prescribed drug. Antibiotic therapy introduction refers to starting antibiotic treatment in a patient who is not on empirical antibiotic therapy. Antibiotic escalation refers to the addition of a new antibiotic or a switch for a broader-spectrum agent. Antibiotic de-escalation refers to the discontinuation of an antibiotic or a switch for a narrower-spectrum agent.

### 5.5. Statistical Analysis

Descriptive data are presented as absolute (n) and relative (%) frequencies. Comparison involving dichotomous variables was tested using the χ2 test or Fisher Exact Test as appropriate. Statistical significance was set at *p*-value < 0.05.

## Figures and Tables

**Table 1 antibiotics-12-00473-t001:** Antimicrobial resistance patterns of *Escherichia coli* blood cultures isolates included in the study.

	CTX-M Positive n = 67 % (n)	CTX-M Negative n = 132 % (n)	*p*-Value
ESBL-producers	100 (67)	3.8 (5)	**<0.01**
AmpC-producers	0	1.5 (2)	0.55
Ceftazidime	76.1 (51)	3 (4)	**<0.01**
Cefotaxime	98.5 (66)	4.6 (6)	**<0.01**
Cefepime	86.6 (58)	2.3 (3)	**<0.01**
Ceftolozane/tazobactam	4.5 (3)	0	**0.04**
Ceftazidime/avibactam	0	0	1
Piperacillin/tazobactam	14.9 (10)	7.6 (10)	0.1
Gentamicin	29.9 (20)	7.6 (10)	**<0.01**
Amikacin	16.4 (11)	0.8 (1)	**<0.01**
Ciprofloxacin	77.6 (52)	18.9 (25)	**<0.01**
Levofloxacin	73.1 (49)	17.4 (23)	**<0.01**
Ertapenem	3 (2)	0	0.11
Meropenem	0	0	1
Imipenem	0	0	1
Colistin	4.5 (3)	0	**0.04**
Sulfamethoxazole/trimethoprim	61.2 (41)	33.3 (44)	**<0.01**

Bold values denote statistical significance at the *p* < 0.05 level. Abbreviation: ESBL: extended-spectrum β-lactamase.

**Table 2 antibiotics-12-00473-t002:** Antimicrobial clinical attitude according to direct detection of CTX-M extended-spectrum-β-lactamase-producers in *Escherichia coli* positive blood cultures.

	CTX-M Positive n = 67 % (n)	CTX-M Negative n = 132 % (n)	*p*-Value
Empirical antibiotic therapy	85.1 (57/67)	91.7 (121/132)	0.15
Combination therapy	15.8 (9/57)	9.9 (12/121)	0.32
3rd-4th-5th generation cephalosporin-containing	15.8 (9/57)	24.8 (30/121)	0.18
Ceftazidime/avibactam-containing	1.8 (1/57)	0	0.30
Amoxicillin/clavulanate-containing	1.8 (1/57)	5 (6/121)	0.67
Piperacillin/tazobactam-containing	45.6 (26/57)	44.6 (54/121)	0.90
Aminoglycoside-containing	15.8 (9/57)	9.9 (12/121)	0.32
Fluoroquinolone-containing	7 (4/57)	5 (6/121)	0.73
Fosfomycin-containing	1.8 (1/57)	1.7 (2/121)	1
Carbapenem-containing	26.3 (15/57)	15.7 (19/121)	0.09
Empirical active antibiotic therapy	73.7 (42/57)	91.7 (111/121)	**<0.01**
Antibiotic therapy modification after direct detection of CTX-M ESBL production result	37.3 (25/67)	13.6 (18/132)	**<0.01**
Antibiotic therapy introduction	24 (6/25)	44.4 (8/18)	0.20
Antibiotic escalation	76 (19/25)	44.4 (8/18)	0.06
Antibiotic de-escalation	0	11.1 (2/18)	0.17
Combination therapy	16 (4/25)	0	0.13
3rd generation cephalosporin-containing	0	33.3 (6/18)	**<0.01**
Ceftazidime/avibactam-containing	8 (2/25)	0	0.50
Piperacillin/tazobactam-containing	4 (1/25)	44.4 (8/18)	**<0.01**
Aminoglycoside-containing	12 (3/25)	11.1 (2/18)	1
Fosfomycin-containing	8 (2/25)	0	0.50
Carbapenem-containing	76 (19/25)	11.1 (2/18)	**<0.01**
Antibiotic therapy modification after antimicrobial susceptibility testing results	28.8 (19/66 *)	20.5 (27/132)	0.19
Antibiotic therapy introduction	15.8 (3/19)	11.1 (3/27)	0.68
Antibiotic escalation	73.7 (14/19)	29.6 (8/27)	**<0.01**
Antibiotic de-escalation	10.5 (2/19)	59.3 (16/27)	**<0.01**
Combination therapy	31.6 (6/19)	7.4 (2/27)	0.05
Amoxicillin/clavulanate-containing	0	14.8 (4/27)	0.13
3rd-4th generation cephalosporin-containing	0	55.6 (15/27)	**<0.01**
Piperacillin/tazobactam-containing	15.8 (3/19)	7.4 (2/27)	0.64
Aminoglycoside-containing	31.6 (6/19)	7.4 (2/27)	0.05
Fluoroquinolone-containing	0	7.4 (2/27)	0.50
Fosfomycin-containing	5.3 (1/19)	0	0.41
Carbapenem-containing	79 (15/19)	7.4 (2/27)	**<0.01**
Sulfamethoxazole/trimethoprim-containing	0	7.4 (2/27)	0.50

Bold values denote statistical significance at the *p* < 0.05 level. * One patient died before being available conventional susceptibility testing results.

## Data Availability

The dataset analyzed during the current study is available from the corresponding author upon reasonable request.

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
