# Peer review of "Impact of NG-Test CTX-M MULTI Immunochromatographic Assay on Antimicrobial Management of Escherichia coli Bloodstream Infections"

_antibiotics, 2023, doi:10.3390/antibiotics12030473_

Round 1
Reviewer 1 Report
1. Abstract. No need for subheadings (background, results…)
2. Introduction. Please merge lines 77-93 into 1 paragraph
3. Line 96-97 and Table 1. Please clarify the descriptions of the method used for ESBL detections, e.g. ____-positive results/___-negative results (PCR? Double disk?)
4. Line 104. 1.5 “%”. Please revise
5. Table 2. Why did only 44 patients receive modified therapies after the detections (even after AST)?
6. Table 2. It was noticed that there were 10 patients in both groups having no empirical therapy. how were the patients treated?
7. It would be better to further analyze and report the patient’s outcome with/without the rapid ESBL test to increase the clinicians’ confidence.
8. Discussion. Please add some other rapid test-related reference
9. Subheadings (4&5). Please put the conclusion right after the discussion and let the material and method be the last section.
10. Conclusion. Please make the description concise. 5-8 lines would be enough
11. Material and methods. Please add a working flowchart to make the experimental design more intuitive.
Author Response
Turin, 14th February 2023
Editor-in-Chief
Antibiotics
We would like to thank the Editorial Team for his helpful suggestions, which in our view have enhanced the quality and strength of our study. We hope that in this revised version the manuscript is now suitable for publication in Antibiotics.
Please, note that the changes to the original manuscript have been highlighted in the text. The response to the Editor’s comments and ensuing modifications in the manuscript are also clearly indicated in the rebuttal.
Comments from Reviewers and point-by-point answers
Reviewer #1:
1) Abstract. No need for subheadings (background, results…)
We would like to thank the Reviewer for this comment. Accordingly, subheadings were eliminated.
2) Introduction. Please merge lines 77-93 into 1 paragraph
We would like to thank the Reviewer for this comment. Accordingly, lines 77-93 were merged into one paragraph.
3) Line 96-97 and Table 1. Please clarify the descriptions of the method used for ESBL detections, e.g. ____-positive results/___-negative results (PCR? Double disk?)
We would like to thank the Reviewer for this comment. One-hundred ninety-nine E. coli BSI events with report of direct detection of CTX-M ESBL production result and confirmed by conventional blood cultures routine were included in the study. EUCAST guidelines were used to identify ESBL-producing E. coli and confirmatory tests for resistance mechanisms were performed once the conventional antimicrobial susceptibility testing results became available. The phenotypic test included in the NMDR micro-dilution panel, based on synergy of β-lactamases inhibitor clavulanic acid on cefotax-ime and ceftazidime MICs, was used to detect ESBLs. Multiplex real-time polymerase chain reaction assay specific for blaCTX-M-like genes (ESBL ELITe MGB Kits, ELITechGroup Molecular Diagnostics, Turin, Italy) was used on isolates that tested positive by phenotypic test for ESBL detection.
4) Line 104. 1.5 “%”. Please revise
We would like to thank the Reviewer for this comment. Accordingly, “%” was added.
5) Table 2. Why did only 44 patients receive modified therapies after the detections (even after AST)?
We would like to thank the Reviewer for this comment. This study was aimed at assessing the impact on antimicrobial prescription of direct detection of CTX-M ESBL-producers in E. coli positive blood cultures. The fact that not all patients changed therapy in accordance with the direct detection of CTX-M ESBL-production result or even after AST may largely mean that the patients were already empirically treated with a therapy considered appropriate (piperacillin/tazobactam- or carbapenem-including regimen). All other considerations that may have had a minor influence are to be referred to the knowledge of the clinical context that has already been made explicit as a limitation of the study.
6) Table 2. It was noticed that there were 10 patients in both groups having no empirical therapy. how were the patients treated?
We would like to thank the Reviewer for this comment. As reported in Table 2, six CTX-M positive and eight CTX-M negative patients started antibiotic therapy after the direct detection of CTX-M ESBL-production, mainly with carbapenem- and piperacillin-tazobactam-including regimens, respectively. Similarly, three patients for both group started antibiotic therapy after AST results. In both scenarios the antibiotic therapy was active. Given the low numbers, we decided not to mention it in the paper.
7) It would be better to further analyze and report the patient’s outcome with/without the rapid ESBL test to increase the clinicians’ confidence.
We would like to thank the Reviewer for this comment. Accepting the indications that the study can provide also means accepting its limitations, which have been well explained. The knowledge of the outcome may only minimally help to increase the clinician's degree of confidence in using the rapid test results as other parameters should be considered (comorbidities, BSI severity etc. etc.). The contribution of the rapid test result together with all the variables just mentioned should be the subject of multivariate analysis or possibly clinical trials, horizons that this study is unable to guarantee.
8) Discussion. Please add some other rapid test-related reference
We would like to thank the Reviewer for this comment. More rapid test-related references were added.
9) Subheadings (4&5). Please put the conclusion right after the discussion and let the material and method be the last section.
We would like to thank the Reviewer for this comment. Accordingly, conclusions were placed before materials and methods.
10) Conclusion. Please make the description concise. 5-8 lines would be enough
We would like to thank the Reviewer for this comment. Accordingly, conclusions were shortened.
11) Material and methods. Please add a working flowchart to make the experimental design more intuitive.
We would like to thank the Reviewer for this comment. This study investigated the clinical impact of a rapid workflow for resistance markers detection applied to blood cultures that has been in place in our laboratory for more than three years. The workflow has already been published (Boattini et al 2021, DOI: 10.1007/s10096-021-04192-8) and mentioned in chapter 5.2 (Rapid blood culture workflow: direct detection of CTX-M ESBL-producing E. coli) and it seems repetitive to have to report it again.

Reviewer 2 Report
Dear authors,
Congrats for the research manuscript!
Thank you!
Author Response
Turin, 14th February 2023
Editor-in-Chief
Antibiotics
We would like to thank the Editorial Team for his helpful suggestions, which in our view have enhanced the quality and strength of our study. We hope that in this revised version the manuscript is now suitable for publication in Antibiotics.
Please, note that the changes to the original manuscript have been highlighted in the text. The response to the Editor’s comments and ensuing modifications in the manuscript are also clearly indicated in the rebuttal.
Comments from Reviewers and point-by-point answers
Reviewer #2:
1) Dear authors, Congrats for the research manuscript! Thank you!
We would like to thank the Reviewer for this enthusiastic comment.

Reviewer 3 Report
I want to appreciate the authors of this manuscript which is aimed at investigating the impact of the direct detection of ESBL on managing antimicrobial resistance. Although the intention of the manuscript is excellent. The poor writing style makes it difficult to comprehend. I'll advise the authors to do a thorough review of the manuscript again.
Author Response
Turin, 14th February 2023
Editor-in-Chief
Antibiotics
We would like to thank the Editorial Team for his helpful suggestions, which in our view have enhanced the quality and strength of our study. We hope that in this revised version the manuscript is now suitable for publication in Antibiotics.
Please, note that the changes to the original manuscript have been highlighted in the text. The response to the Editor’s comments and ensuing modifications in the manuscript are also clearly indicated in the rebuttal.
Comments from Reviewers and point-by-point answers
Reviewer #3:
1) I want to appreciate the authors of this manuscript which is aimed at investigating the impact of the direct detection of ESBL on managing antimicrobial resistance. Although the intention of the manuscript is excellent. The poor writing style makes it difficult to comprehend. I'll advise the authors to do a thorough review of the manuscript again.
We would like to thank the reviewer for these accurate appraisals. Accordingly, the writing style has been largely revised.

Reviewer 4 Report
This study assessed the impact of rapid detection of CTX-M extended spectrum beta-lactamase E. coli from bloodstream infections, based on lateral flow immune assays (NG-Test CTX-M MULTI assay from NG Biotech). They analysed almost 200 cases and almost 1/3 of them were CTX-M ESBL positive. Rapid detection allowed for a significantly better treatment and optimization of antibiotic selection, on the same day of BC processing.
The paper was well written and easy to follow and the results were transparent, well presented in tables.
I would recommend that the authors add a section in Discussion, on the most modern approaches concerning antimicrobial resistance prediction with Next Generation whole genome Sequencing. The authors should mention that a relatively small number of Pathogenic Bacteria Are of High Importance and Impact, with E.coli being the most clinically prevalent and the most sequenced pathogen (DOI:10.3390/microorganisms10051040). Reference laboratories of Public Health England and the Scottish Healthcare Associated Infection Prevention Institute have adopted WGS as a routine method to analyse samples from bacterial pathogens including E. coli and Shigella (DOI: 10.1016/j.jhin.2020.11.001). WGS has been evaluated against traditional phenotypic approaches, in order to determine AMR profiles with mixed or even poor results initially. However, more recent studies have now demonstrated very high levels of concordance (DOI: 10.1093/jac/dkaa345). The authors should include all the above most modern approaches and discuss that although their approach is not as sophisticated, nevertheless, it is a cheap, fast, not as laborious intensive and impactful approach that constitutes an improvement, until the most modern WGS approaches become standard in clinical practice.
Some minor corrections:
Line 16: events
Line 61: ESBL enzymes
Line 61-64: This sentence is rather long, better break it in two.
Line 67-69: Please rephrase.
Line 104: 1.5%
Line 129: in a hospital
Author Response
Turin, 14th February 2023
Editor-in-Chief
Antibiotics
We would like to thank the Editorial Team for his helpful suggestions, which in our view have enhanced the quality and strength of our study. We hope that in this revised version the manuscript is now suitable for publication in Antibiotics.
Please, note that the changes to the original manuscript have been highlighted in the text. The response to the Editor’s comments and ensuing modifications in the manuscript are also clearly indicated in the rebuttal.
Comments from Reviewers and point-by-point answers
Reviewer #4:
1) This study assessed the impact of rapid detection of CTX-M extended spectrum beta-lactamase E. coli from bloodstream infections, based on lateral flow immune assays (NG-Test CTX-M MULTI assay from NG Biotech). They analysed almost 200 cases and almost 1/3 of them were CTX-M ESBL positive. Rapid detection allowed for a significantly better treatment and optimization of antibiotic selection, on the same day of BC processing. The paper was well written and easy to follow and the results were transparent, well presented in tables.
We would like to thank the Reviewer for these accurate appraisals.
2) I would recommend that the authors add a section in Discussion, on the most modern approaches concerning antimicrobial resistance prediction with Next Generation whole genome Sequencing. The authors should mention that a relatively small number of Pathogenic Bacteria Are of High Importance and Impact, with E.coli being the most clinically prevalent and the most sequenced pathogen (DOI:10.3390/microorganisms10051040). Reference laboratories of Public Health England and the Scottish Healthcare Associated Infection Prevention Institute have adopted WGS as a routine method to analyse samples from bacterial pathogens including E. coli and Shigella (DOI: 10.1016/j.jhin.2020.11.001). WGS has been evaluated against traditional phenotypic approaches, in order to determine AMR profiles with mixed or even poor results initially. However, more recent studies have now demonstrated very high levels of concordance (DOI: 10.1093/jac/dkaa345). The authors should include all the above most modern approaches and discuss that although their approach is not as sophisticated, nevertheless, it is a cheap, fast, not as laborious intensive and impactful approach that constitutes an improvement, until the most modern WGS approaches become standard in clinical practice.
We would like to thank the Reviewer for these accurate appraisals. Accordingly, discussion was largely revised.
3) Some minor corrections:
Line 16: events
Line 61: ESBL enzymes
Line 61-64: This sentence is rather long, better break it in two.
Line 67-69: Please rephrase.
Line 104: 1.5%
Line 129: in a hospital
We would like to thank the Reviewer for these comments. Accordingly, all typing errors have been corrected and sentences rephrased.

Round 2
Reviewer 3 Report
Authors need to make some English editing to the manuscript